# Ensemble learning for Physics Informed Neural Networks: a Gradient Boosting approach

**Zhiwei Fang**
`pentilm@outlook.com`

**Sifan Wang**
Graduate Group in Applied Mathematics
and Computational Science
University of Pennsylvania
Philadelphia, PA 19104
`sifanw@sas.upenn.edu`

**Paris Perdikaris**
Department of Mechanichal Engineering
and Applied Mechanics
University of Pennsylvania
Philadelphia, PA 19104
`pgp@seas.upenn.edu`

## Abstract

While the popularity of physics-informed neural networks (PINNs) is steadily rising, to this date, conventional PINNs have not been successful in simulating multi-scale and singular perturbation problems. In this work, we present a new training paradigm referred to as "gradient boosting" (GB), which significantly enhances the performance of physics informed neural networks (PINNs). Rather than learning the solution of a given partial differential equation (PDE) using a single neural network directly, our algorithm employs a sequence of neural networks to achieve a superior outcome. This approach allows us to solve problems presenting great challenges for traditional PINNs. Our numerical experiments demonstrate the effectiveness of our algorithm through various benchmarks, including comparisons with finite element methods and PINNs. Furthermore, this work also unlocks the door to employing ensemble learning techniques in PINNs, providing opportunities for further improvement in solving PDEs.

## 1 Introduction

Physics informed neural networks have recently emerged as an alternative to traditional numerical solvers for simulations in fluids mechanics Raissi et al. (2020); Sun et al. (2020), and bio-engineering Sahli Costabal et al. (2020); Kissas et al. (2020). However, PINNs using fully connected, or some variants architectures such as Fourier feature networks Tancik et al. (2020), fail to accomplish stable training convergence and yield accurate predictions at whiles, especially when the underlying PDE solutions exhibit high-frequencies or multi-scale features Fuks & Tchelepi (2020); Raissi (2018). To mitigate this pathology, Krishnapriyan et al. (2021) proposed a sequence-to-sequence learning method for time-dependent problems, which divide the time domain into sub-intervals and solve the problem progressively on each them. This method avoids the pollution of the underlying solution due to the temporal error accumulation. Wang et al. (2022) elaborated the reason that the PINNs fail to train from a neural tangent kernel perspective, and proposed an adaptive training strategy to improve the PINNs' performance. Although all of these works were demonstrated to produce significant and consistent improvements in the stability and accuracy of PINNs, the fundamental reasons behind the practical difficulties of training fully-connected PINNs still remain unclear Fuks & Tchelepi (2020).

Besides PINNs, many other machine learning tasks suffer from the same issues, and some of these issues have been resolved by gradient boosting method. The idea of gradient boosting is blending several weak learners into a fortified one that gives better predictive performance than could be obtained from any of the constituent learners alone Opitz & Maclin (1999). For example, Zhang &

Haghani (2015) proposes a gradient-boosting tree-based travel time prediction method, driven by the successful application of random forest in traffic parameter prediction, to uncover hidden patterns in travel time data to enhance the accuracy and interpretability of the model. Recently, many researchers have contributed to gradient boosting method and further improved its performance. Friedman (2002) shows that both the approximation accuracy and execution speed of gradient boosting can be substantially improved by incorporating randomization into the procedure, and this randomized approach also increases robustness against overcapacity of the base learner.

Inspired by the above-mentioned literature review, we arrive at our proposed method in this paper. In this work, we present a gradient boosting physics informed neural networks (GB PINNs), which adopts a gradient boosting idea to approximate the underlying solution by a sequence of neural networks and train the PINNs progressively. Specifically, our main contributions can be summarized into the following points:

1. Inspired by the GB technique prevalent in traditional machine learning, we introduce a GB PINNs approach, improving the training convergence and approximation to sharp gradients

2. Our method's enhanced performance is justified from various perspectives. Initially, drawing parallels with traditional gradient boosting, we empirically show that a composite learner outperforms isolated ones.

3. To substantiate our techniques, we present several rigorous experiments, encompassing even those in numerical analysis. We provide an ablation study and detail the trade-offs between time and memory, aiding readers in comprehending the practical implications of our method.

We present our algorithm with motives in section 2. Numerical experiments are shown in section 3 to verify our algorithm. We discuss our algorithm and conclude the paper in section 4. Some of the details about the algorithm and the numerical results can be found in appendix A.

## 2 GRADIENT BOOSTING PHYSICS INFORMED NEURAL NETWORKS

In this section, we will introduce our algorithm directly. We put the motivation of our algorithm in appendix A.2. Let us first introduce some notations. Let $f(\boldsymbol{x}; \theta)$ be a neural network with input $\boldsymbol{x}$ and parameterized by $\theta$. For a sequence of neural networks with parameters $\theta_0, \theta_1, \cdots, \theta_m$, define $\Theta_m = \cup_{i=0}^{m} \{\theta_i\}$. The proposed algorithm in this paper can be summarized as follows:

---

**Algorithm 1** Gradient boosting physics informed neural network.

---

**Require:**
    A baseline neural network $f_0(\boldsymbol{x}; \theta_0)$ and an ordered neural network set $\{h_m(\boldsymbol{x}; \theta_m)\}_{m=1}^{M}$ that contains models going to be trained in sequence;
    A set of learning rate $\{\rho_m\}_{m=0}^{M}$ that correspond to $\{f_0(\boldsymbol{x}; \theta_0)\} \cup \{h_m(\boldsymbol{x}; \theta_m)\}_{m=1}^{M}$. Usually, $\rho_0 = 1$ and $\rho_m$ is decreasing in $m$;
    Set $f_m(\boldsymbol{x}; \Theta_m) = f_{m-1}(\boldsymbol{x}; \Theta_{m-1}) + \rho_m h_m(\boldsymbol{x}; \theta_m)$,     for $m = 1, 2, 3, \cdots, M$.
    Given PDEs problem, establish the corresponding loss by theory of PINNs.
**Ensure:**
  1: Train $f_0(\boldsymbol{x}; \theta_0) = \rho_0 f_0(\boldsymbol{x}; \theta_0)$ to minimize the PINNs' loss.
  2: **for** $m = 1$ to **M do**
  3:     In $f_m(\boldsymbol{x}; \Theta_m) = f_{m-1}(\boldsymbol{x}; \Theta_{m-1}) + \rho_m h_m(\boldsymbol{x}; \theta_m)$, set trainable parameters as $\theta_m$. Train $f_m(\boldsymbol{x}; \Theta_m)$ to minimize PINNs' loss.
  4: **end for**
  5: **return** $f_M(\boldsymbol{x}; \Theta_M)$ as a predictor of the solution of the PDE problem.

---

The proposed algorithm, described in Algorithm 1, utilizes a sequence of neural networks and an iterative update procedure to minimize the loss gradually. At each iteration step $i$, the forward prediction relies on the union parameter set $\Theta_i$, while the backward gradient propagation is only performed on $\theta_i$. This results in a mild increase in computational cost during the training of GB iteration. The simplicity of this algorithm allows practitioners to easily transfer their PINNs codes to GB PINNs'. In the following section, we will demonstrate that this simple technique can enable PINNs to solve many problems that present challenges for the original formulation of Raissi et al. (2019).

## 3 NUMERICAL EXPERIMENTS

In this section, we will demonstrate the effectiveness of the proposed GB PINNs algorithm through a comprehensive set of numerical experiments. To simplify the notation, we use a tuple of numbers to denote the neural network architecture, where the tuple represents the depth and width of the layers. For example, a neural network with a two-dimensional input and a one-dimensional output, as well as two hidden layers with width 100 is represented as $(2, 100, 100, 1)$. The set up of the experiments can be found in appendix A.3. To quantify the model's accuracy, we use the relative $l^2$ error over a set of points $\{x_i\}_{i=1}^N$:

$$\text{Error} = \frac{\sum_{i=1}^N |u_{pred}(x_i) - u_{true}(x_i)|^2}{\sum_{i=1}^N |u_{true}(x_i)|^2},$$

where $u_{pred}$ and $u_{true}$ are the prediction and ground truth of the solution for a given PDE, respectively.

In our analysis, we assess the error across a defined set of grid points. In the subsequent experiments, we produce a set of $1,000$ equidistant points for each dimension within the domain. These points are then combined using the Cartesian product to establish the grid coordinates.

We also clarify that all the predictions of numerical experiments below are in-domain. That is, the training data and test data are coming from the same domain.

### 3.1 1D SINGULAR PERTURBATION

In this first example, we utilize GB PINNs to solve the following 1D singular perturbation problem.

$$-\varepsilon^2 u''(x) + u(x) = 1, \qquad \text{for } x \in (0, 1),$$
$$u(0) = u(1) = 0.$$

The perturbation parameter, $0 < \varepsilon \ll 1$, is set to $10^{-4}$ in this case. The exact solution to this problem is given by

$$u(x) = 1 - \frac{e^{-x/\varepsilon} + e^{(x-1)/\varepsilon}}{1 + e^{-1/\varepsilon}}.$$

Despite the boundedness of the solution, it develops boundary layers at $x = 0$ and $x = 1$ for small values of $\varepsilon$, a scenario in which traditional PINNs have been known to perform poorly.

Utilizing the notation established in Algorithm 1, we designate $f_0$ as $(1, 50, 1)$, $h_1$ as $(1, 100, 1)$, $h_2$ as $(1, 100, 100, 1)$, $h_3$ as $(1, 100, 100, 100, 1)$ and $h_4$ as a Fourier feature neural network $(1, 50, 50, 1)$ with frequencies ranging from 1 to 10. For the sake of clarity in subsequent examples, we will simply present a series of network architectures, which will implicitly represent the $f_0$ and $h_m$ ($m = 0, 1, 2, \cdots$) configurations in sequence. The details of the Fourier feature method used in this study can be found in the appendix A.1. For each GB iteration, we train $10,000$ steps using a dataset of $10,000$ uniform random points in $(0, 1)$.

The relative $l^2$ error is found to be $0.43\%$ for GB PINNs. As a comparison, we use comparable vanilla PINNs but the relative error is $12.56\%$. More detail can be found in appendix A.4.

It is worth mentioning that, as demonstrated in Appendix A.4, the mere incorporation of Fourier features can significantly enhance accuracy. This observation holds true for the subsequent two examples presented in our study. Nevertheless, our proposed methodology, which amalgamates traditional PINNs with Fourier features, consistently yields superior results.

### 3.2 2D SINGULAR PERTURBATION WITH BOUNDARY LAYERS

In this example, we aim to solve the Eriksson-Johnson problem, which is a 2D convection-dominated diffusion equation. As previously noted in the literature, such as in Demkowicz & Heuer (2013), this problem necessitates the use of specialized finite element techniques in order to obtain accurate solutions, such as the Discontinuous Petrov Galerkin (DPG) finite element method.

Let $\Omega = (0, 1)^2$. The model problem is

$$-\varepsilon \Delta u + \frac{\partial u}{\partial x} = 0 \qquad \text{in } \Omega,$$
$$u = u_b \qquad \text{on } \partial\Omega.$$

The manufactured solution is

$$u(x, y) = \frac{e^{r_1(x-1)} - e^{r_2(x-1)}}{e^{-r_1} - e^{-r_2}} \sin(\pi y) \quad \text{with} \quad r_{1,2} = \frac{-1 \pm \sqrt{1 + 4\varepsilon^2\pi^2}}{-2\varepsilon}.$$

In this example, we set $\varepsilon = 10^{-3}$. To resolve this problem, we sequentially employ a range of neural network architectures as follows: $(2, 50, 1)$, $(2, 100, 1)$, $(2, 100, 100, 1)$, $(2, 100, 100, 100, 1)$, $(2, 100, 100, 1)$, culminating in a Fourier feature network $(1, 100, 100, 1)$ with frequencies ranging from 1 to 50. All the other set up are kept the same as the previous experiment. The GB PINNs' relative l2 and vanilla PINNs' are $1.03\%$ and $57.66\%$, respectively. More detail can be found in appendix A.5.

### 3.3 2D SINGULAR PERTURBATION WITH AN INTERIOR BOUNDARY LAYER

In this example, we address a 2D convection-dominated diffusion problem featuring curved streamlines and an interior boundary layer. The model problem is

$$\begin{aligned} -\varepsilon\Delta u + \beta \cdot \nabla u &= f & \text{in } \Omega, \\ u &= u_b & \text{on } \partial\Omega, \end{aligned}$$

with $\beta = e^x(\sin(y), \cos(y))$ and $f$, $u_0$ are defined such that

$$u(x, y) = \arctan\left(\frac{1 - \sqrt{x^2 + y^2}}{\varepsilon}\right).$$

This example has been solved by DPG finite element method in Demkowicz & Heuer (2013). A specific value of $\epsilon = 10^{-4}$ was chosen for the purpose of this study. The neural network architectures are sequentially employed in the following order: $(2, 200, 200, 200, 1)$, $(2, 100, 100, 100, 1)$, $(2, 100, 100, 1)$, culminating in a Fourier feature network $(2, 50, 50, 1)$ with frequency ranging from 1 to 5. The relative $l^2$ error for GB PINNs and vanilla PINNs are $3.37\%$ and $43\%$, respectively. More detail can be found in appendix A.6.

### 3.4 2D NONLINEAR REACTION-DIFFUSION EQUATION

In this example, we investigate the solution of a time-dependent nonlinear reaction-diffusion equation. As demonstrated in Krishnapriyan et al. (2021), conventional PINNs have been shown to be inadequate in accurately learning the solution of such equations.

Let $\Omega = (0, 2\pi)$. The model problem is

$$\begin{aligned} \frac{\partial u}{\partial t} - 10\frac{\partial^2 u}{\partial x^2} - 6u(1-u) &= 0, & x \in \Omega, t \in (0, 1], \\ u(x, 0) &= h(x) & x \in \Omega, \end{aligned}$$

with periodic boundary conditions, where

$$h(x) = e^{-\frac{(x-\pi)^2}{2(\pi/4)^2}}.$$

In order to impose an exact periodic boundary condition, we use $(\sin(x), \cos(x))$ as the spatial input instead of $x$, while maintaining the temporal input unchanged. This eliminates the need for boundary loss. For this problem, we employ neural network architectures in the following sequential order: $(2, 200, 200, 200, 1)$, $(2, 100, 100, 100, 1)$, $(2, 100, 100, 1)$. The relative $l^2$ error of GB PINNs is $0.58\%$, while the error reported in Krishnapriyan et al. (2021) is about $50\%$. More detail can be found in A.7.

To conclude this section, we further elaborate on the comparative analysis of our proposed method against various other variants of PINNs, as detailed in Section A.8.

## 4 CONCLUSION

In this paper, we propose a GB PINNs algorithm, which utilizes multiple neural networks in sequence to predict solutions of PDEs. The algorithm is straightforward to implement and does not require

extensive fine-tuning of neural network architectures. Additionally, the method is flexible and can be easily integrated with other PINNs techniques. Our experimental results demonstrate its effectiveness in solving a wide range of PDE problems with a special focus on singular perturbation.

However, it should be noted that the algorithm has some limitations. Firstly, it is not suitable for solving conservation laws with derivative blow-ups, such as the inviscid Burgers' equation and the Sod shock tube problem. This is due to the lack of sensitivity of these equations' solutions to PDE loss. The addition of more neural networks alone cannot overcome this issue. Secondly, the optimal combination of neural networks is not always clear, and the current experimental selection is mostly based on experience and prior estimation of the PDE problem. Further research into the theoretical and quantitative analysis of this method is an interesting direction for future work.

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

# A    APPENDIX

## A.1    FOURIER FEATURES NETWORKS

In this subsection, we present the Fourier feature network structure that we employed in our experiments. For a more in-depth explanation of Fourier features, please refer to Tancik et al. (2020).

Given an input $x \in \mathbb{R}^{N \times d}$ for the neural network, where $N$ is the batch size and $d$ is the number of features, we encode it as $v = \gamma(x) = [\cos(2\pi x B), \sin(2\pi x B)] \in \mathbb{R}^{N \times 2m}$, where $B \in \mathbb{R}^{d \times m}$, and $m$ is half of the output dimension of this layer. The encoded input $v$ is then passed as input to the subsequent hidden layers, while the rest of the neural network architecture remains the same as in a traditional MLP. Different choices of $B$ result in different types of Fourier features.

In our experiment, we utilized the axis Fourier feature. We first selected a range of integers as our frequencies and denoted them as $f = (f_0, f_1, \cdots, f_p) \in \mathbb{R}^p$. We then constructed a block matrix $B = [F_0, F_1, \cdots, F_p] \in \mathbb{R}^{d \times dp}$, where $F_i$ is a diagonal matrix with all of its diagonal elements set to $f_i$. This results in the desired matrix $B \in \mathbb{R}^{d \times dp}$, and $dp = m$ is half of the output dimension for this layer.

## A.2    GB PINN ALGORITHM MOTIVATION

Despite a series of promising results in the literature Hennigh et al. (2021); Kissas et al. (2020); Raissi (2018); Raissi et al. (2020); Sun et al. (2020), the original formulation of PINNs proposed by Raissi et al. (2019) has been found to struggle in constructing an accurate approximation of the exact latent solution. While the underlying reasons remain largely elusive in general, certain failure modes have been explored and addressed, as evidenced in Wang et al. (2021; 2022); Krishnapriyan et al. (2021). However, some observations in the literature can be used to infer potential solutions to this issue. One such observation is that the prediction error in PINNs is often of high frequency, with small and intricate structures, as seen in figures 4(b) and 6(a) and (b) of Wang et al. (2022). As demonstrated in Tancik et al. (2020), high-frequency functions can be learned relatively easily using Fourier features. Based on these findings, it is natural to consider using a multi-layer perceptrons (MLPs) as a baseline structure in PINNs, followed by a Fourier feature network, to further minimize the error. This idea led to the development of GB PINNs.

Table 1: Default Experiment Set up

| Name | Value |
|---|---|
| Activation function | GeLU |
| Method to initialize the neural network | Xavier |
| Optimizer | Adam |
| learning rate | $10^{-3}$ |
| learning rate decay period | $10,000$ |
| learning rate decay rate | $0.95$ |

GB is an innovative algorithm designed to enhance the predictive capability of a model by integrating multiple weak learners to form a more robust, strong learner. This process necessitates the computation of derivatives of the loss function with respect to the model function. However, in the realm of PINNs, the loss function typically incorporates spatial or temporal derivatives, which complicates the direct application of GB. To circumvent this issue, our approach modifies the standard procedure. Instead of applying GB directly, we augment the baseline model with an additional model. This new model is trained using the same dataset and loss function as the original. During this process, all parameters from the pre-existing models are kept constant (frozen), meaning they contribute to the forward prediction phase but are excluded from the backward propagation phase. Additionally, we have implemented an empirical learning rate, designed to decay exponentially as the model converges towards the minimum loss. This adjustment ensures a more efficient and stable training process, especially in the context of complex derivative computations inherent to PINNs.

Numerous studies have addressed singular perturbation problems in the context of advection equations, including notable works like those of Deng et al. (2023) and Frerichs-Mihov et al.. However, our approach differs significantly as we do not leverage specific properties of advection equations. Instead, we tackle the problem through the direct application of GB PINN. This method exhibits a higher degree of generality, making it applicable to a broader range of scenarios, including time-dependent nonlinear problems, as demonstrated by our numerical examples.

## A.3 NUMERICAL EXPERIMENTS SETUP

Our default experimental setup is summarized in Table 1, and will be used in all experiments unless otherwise specified.

## A.4 RESULTS OF 1D SINGULAR PERTURBATION

The output of GB PINNs is shown in Figure 1, where the relative $l^2$ error is found to be $0.43\%$.

The boundary layers at $x = 0$ and $x = 1$ are clearly visible in the solution, which is a result of the thinness of the layers and the almost right angle curvature of the solution at these points. Despite the singularity present in the solution, GB PINNs were able to provide an accurate solution for this problem. To further highlight the contribution of GB PINNs in this example, an ablation study was conducted. A vanilla PINNs approach, using a network structure of $(1, 256, 256, 256, 256, 1)$ and $50,000$ training steps, was used to solve the same problem. To make this comparison fair, we use the same amount of training points in total. Therefore, we set the batch size of the training points as the same as before. The training takes $249.47$s. Notably, even though this network possesses greater depth and width than any single network in the GB PINNs ensemble, the resulting relative $l^2$ error is a much higher $12.56\%$, as shown in Figure 2. Additional experiments including ablation studies and comparisons can be found in appendix 2.

However, getting this high level of accuracy comes with its costs. Using a series of neural networks instead of just one means more time and memory are needed. The training times for the GB PINNs networks are $57.44$s, $77.63$s, $126.96$s, $196.08$s, and $259.44$s, respectively. The peak memory requisition touched $0.28$GB. In a scenario where only the largest configuration within the GB PINNs' network spectrum, specifically $(1, 100, 100, 100, 1)$, is utilized, the memory footprint scales down to $0.16$GB. Compared to the standard PINNs, this method takes about three times longer and uses twice the memory. But the accuracy was more than 20 times better. Another discernible trend is the

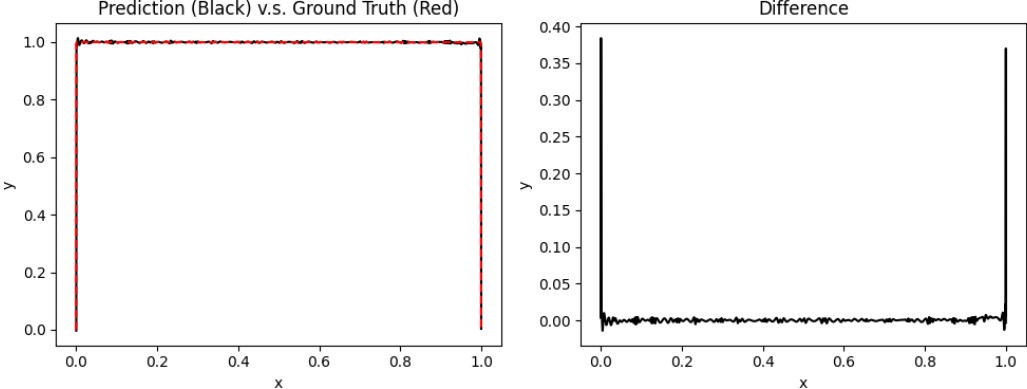

Figure 1: Prediction of singular perturbation problem by GB PINNs, $\varepsilon = 10^{-4}$. Left: predicted solution (black) v.s. ground truth (red). Right: pointwise error.

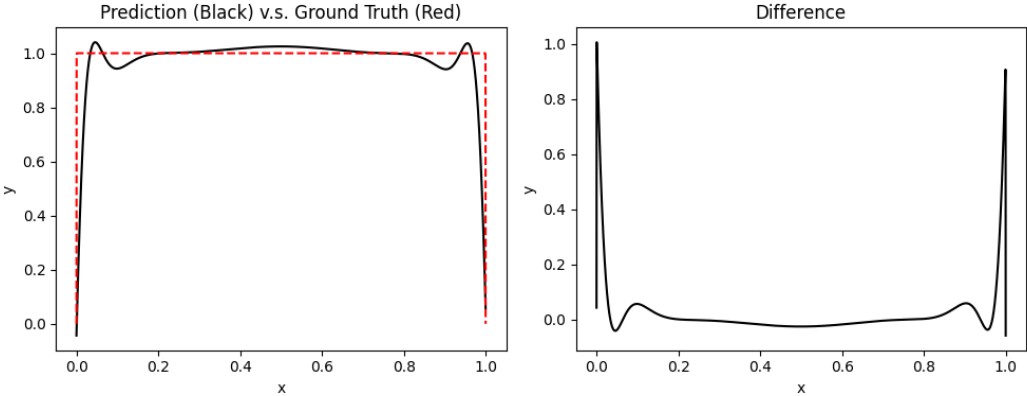

Figure 2: Prediction of singular perturbation problem by PINNs for ablation study, $\varepsilon = 10^{-4}$. Left: predicted solution (black) v.s. ground truth (red). Right: pointwise error.

incremental rise in training time in correlation to the network's order. As we advance to training the $m$-th network, even with the parameters of the preceding networks remaining static, the computational intensity during both forward and backward propagations escalates, leading to protracted training durations.

## A.5  RESULTS OF 2D SINGULAR PERTURBATION WITH BOUNDARY LAYERS

For each iteration of our GB algorithm, we train for $20,000$ steps. The batch sizes for PDE residuals and boundaries are set at $10,000$ and $50,00$, respectively. The predicted solution is visualized in Figure 3. We can see that our model prediction is in good agreement with the ground truth, with a relative $l^2$ error of $1.03\%$. The training times of each individual network are 92.72s, 119.62s, 204.66s, 329.35s, and 459.93s, respectively. The maximum memory consumption reached 0.71GB. However, when only using the largest network configuration, $(2, 100, 100, 100, 1)$, the peak memory usage stands at 0.31GB.

Notably, there is a boundary layer present on the right side of the boundary ($x = 1$), which is not easily recognizable to the naked eye due to its thinness. However, GB PINNs are able to provide a reasonable degree of predictive accuracy even in this challenging scenario.

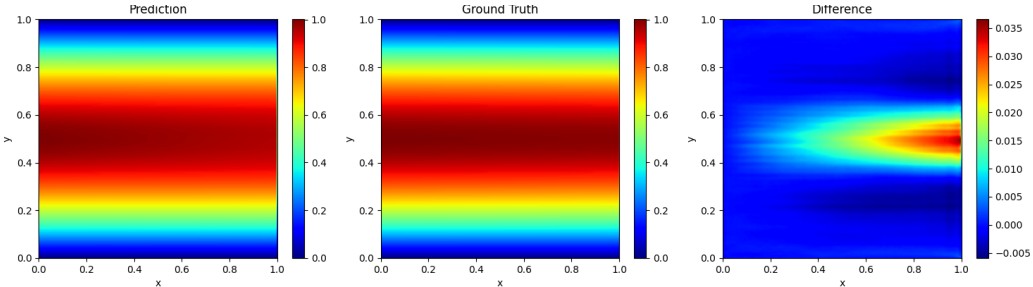

Figure 3: Prediction of 2D singular perturbation with boundary problem by GB PINNs, $\varepsilon = 10^{-3}$. Left: predicted solution. Middle: ground truth. Right: pointwise error.

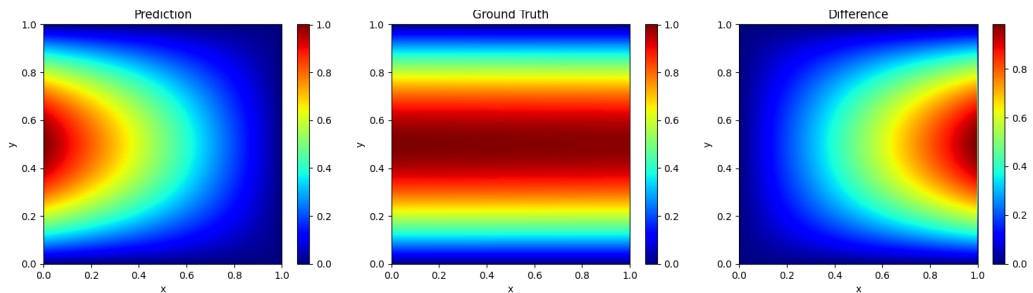

Figure 4: Prediction of 2D singular perturbation with boundary problem by PINNs, $\varepsilon = 10^{-3}$. Left: predicted solution. Middle: ground truth. Right: pointwise error.

To further demonstrate the efficacy of our proposed method, we also attempted to solve this problem using a single fully connected neural network of architecture $(2, 256, 256, 256, 256, 1)$. We train this network by $20,000$ steps under the same hyperparameter settings as before. For a fair comparison, we maintain the same total number of training points. Consequently, we have increased the batch size of the training points by fivefold. However, the resulting relative $l^2$ error was $57.66\%$. This training consumed $811.82$s of time. As can be seen in Figure 4, there is a significant discrepancy between the predicted solution and the reference solution. Additional experimental results, including an ablation study and comparisons, can be found in Appendix 3.

### A.6   RESULTS OF 2D SINGULAR PERTURBATION WITH AN INTERIOR BOUNDARY LAYER

The batch size for the PDE residual and boundary were set to $10,000$ and $5,000$, respectively. For each iteration of our GB algorithm, we train for $20,000$ steps. The results of this study are shown in Figure 5 and exhibit a relative $l^2$ error of $3.37\%$. The training times for each individual network are $409.65$s, $463.84$s, $525.87$s, and $562.61$s, respectively. The peak memory allocation for the GB PINNs and the largest configuration are $0.84$GB and $0.63$GB, respectively.

As a part of an ablation study, we resolve this problem using a fully connected neural network architecture of $(2, 512, 512, 512, 512, 1)$, while maintaining the other configurations as same as the previous experiment. We train this network by $20,000$ steps. For a fair comparison, we maintain the same total number of training points. Consequently, we have increased the batch size of the training points by fourfold. This training consumed $1067.96$s of time. The relative $l^2$ error obtained in this case is $43\%$. The predictions and the corresponding errors are depicted in Figure 6. Additional experiments pertaining to the ablation study and comparisons can be found in the appendix, section 4.

### A.7   RESULTS OF 2D NONLINEAR REACTION-DIFFUSION EQUATION

The weights for the PDE residual and initial condition loss are set to $1$ and $1,000$, respectively. The batch sizes for the PDE residual and initial condition loss are $20,000$ and $1,000$, respectively. For

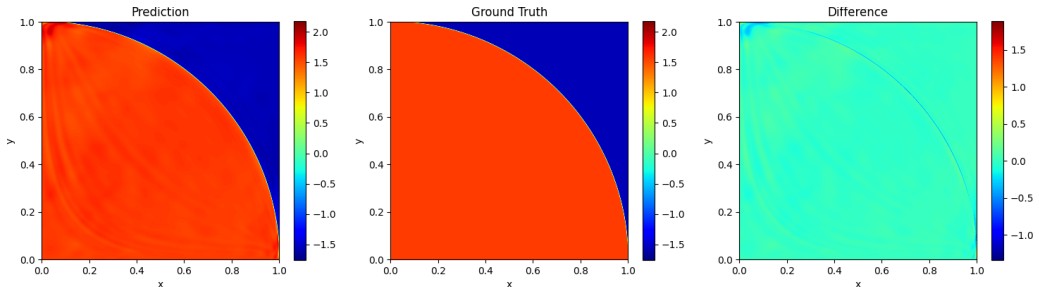

Figure 5: Prediction of 2D singular perturbation with interior boundary problem by GB PINNs, $\varepsilon = 10^{-4}$. Left: predicted solution. Middle: ground truth. Right: pointwise error.

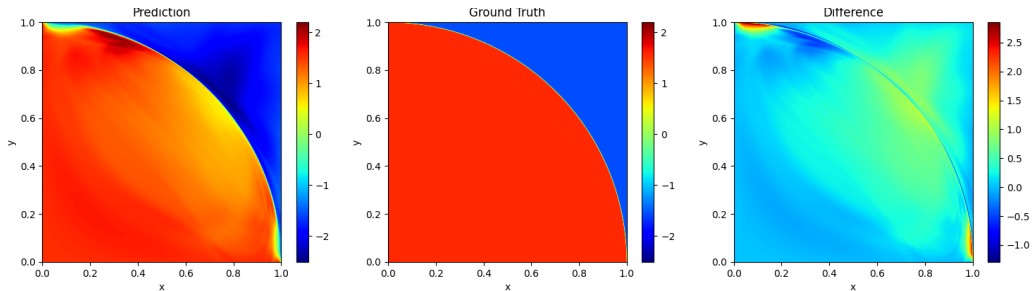

Figure 6: Prediction of 2D singular perturbation with interior boundary problem by PINNs, $\varepsilon = 10^{-4}$. Left: predicted solution. Middle: ground truth. Right: pointwise error.

each iteration of our GB algorithm, we train for $20,000$ steps. We present our results in Figure 7. The relative $l^2$ error is $0.58\%$. As shown in Krishnapriyan et al. (2021), the relative error for traditional PINNs with $\rho = \nu = 5$ is $50\%$. A comparison between the exact solution and the PINNs' prediction can also be found in the figure. The training times for each individual network are $391.99$s, $483.21$s, and $579.77$s, respectively. The peak memory allocation for the GB PINNs and the largest configuration are $0.99$GB and $0.77$GB, respectively.

In the aforementioned study, Krishnapriyan et al. (2021) proposed a sequence-to-sequence learning approach to address this problem, achieving a relative $l^2$ error of $2.36\%$ for $\rho = \nu = 5$. This approach begins by uniformly discretizing the temporal domain, resulting in sequential subintervals. Each of these temporal subintervals is then combined with the spatial domain, forming distinct subdomains. The problem is solved sequentially by applying traditional PINNs through these spatio-temporal

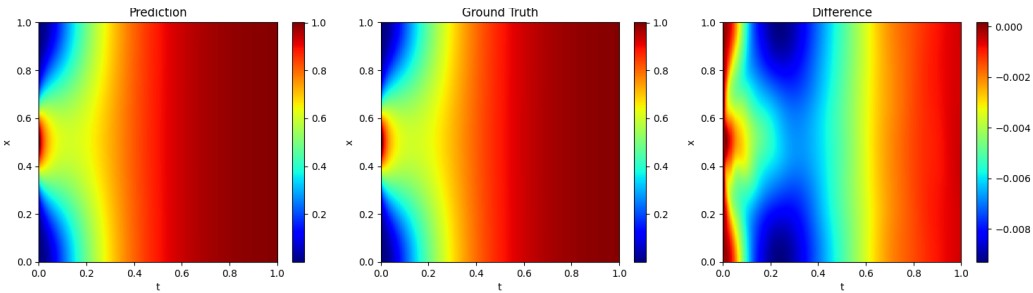

Figure 7: Prediction of nonlinear reaction-diffusion equation by GB PINNs. Left: predicted solution. Middle: ground truth. Right: pointwise error.

subdomains. For the cited example above, the sequence-to-sequence model trained the neural network across 20 such intervals, necessitating the solution of 20 consecutive problems via PINNs.

In contrast, our methodology, executed with $\nu = 10$ and $\rho = 6$, which is a notably more complex setting, required training for only three networks. Rather than partitioning the domain at the physical level and employing multiple learners to construct solutions on subdomains before amalgamating them, our strategy utilizes supplemental learners to reinforce the base learner, thereby enhancing its precision. Remarkably, our approach yielded an error rate nearly four times lower than the former method, signifying a substantial enhancement.

### A.8 COMPARISON WITH OTHER VARIANTS OF PINNS

The numerical examples provided clearly illustrate the significant improvements our proposed method offers over traditional PINNs. Despite the advancements, it is acknowledged that numerous other PINN variants have been developed to enhance performance. It is natural for readers to be curious about how these competitive methods stack up. While it is not feasible to cover all variants exhaustively, we have selected a few that either employ multiple networks or share similarities with our approach for a theoretical comparison.

The study by Arzani et al. (2023) introduces a structure influenced by perturbation theory, where separate networks are designated for inner and outer regions and integrated using gauge functions. This approach, akin to ours, utilizes multiple networks; however, these networks operate across distinct domain regions, requiring the decomposition of the original problem into inner and outer problems. Unlike this method, our approach does not necessitate any mathematical decomposition, presenting a notable advantage, especially as it is not confined to addressing perturbation issues alone.

Jagtap & Karniadakis (2020) unveiled the Extended PINNs (XPINNs), which parallels the earlier mentioned methodology. XPINNs segment the domain into multiple subdomains, each with its dedicated network, and the solutions are unified through the overlap between these subdomains. Despite necessitating some preparatory steps, XPINNs have been shown to be particularly effective for domains with large and complex geometries. While literature specifically addressing XPINNs' application to singular perturbation problems is scarce, we posit that they could either alleviate or simplify these challenges. XPINNs boast the advantage of parallel network training, in contrast to our method's sequential training requirement. However, a potential drawback of XPINNs is the need for local point generators within each subdomain, with performance heavily dependent on how subdomain overlaps are managed.

Our analysis also includes a comparison with the work of Krishnapriyan et al. (2021), which utilizes a sequence-to-sequence approach akin to a temporal domain decomposition. Despite demonstrating lower accuracy, as highlighted in our results, this method necessitates the use of multiple networks and sequential training. In their implementation, the domain was divided into 10 subdomains, significantly increasing the computational effort. This comparison underscores the efficiency and effectiveness of our proposed method.

### A.9 ADDITIONAL ABLATION STUDY RESULTS

In this subsection, we present additional experiments to demonstrate the relative $l^2$ error for various neural network selections. The objective of these experiments is to demonstrate the following:

1. The use of GB PINNs can significantly improve the accuracy of a single neural network.
2. The method is relatively insensitive to the selection of neural networks, as long as the spectral and capacity are sufficient.

In order to simplify the presentation, we will only display the hidden layers of fully-connected neural networks and utilize the list notation in Python. For instance, $[50] * 3$ represents three hidden layers, each containing 50 neurons. The input and output layers are implied by context and not explicitly shown. For Fourier feature neural networks, we denote them using the notation $F_k$, where $k$ represents the range of frequencies used (e.g., $F_{10}[100] * 2$ denotes a Fourier feature neural network with frequencies ranging from 1 to 10, and two hidden layers with 100 neurons each). In the case of the 2D nonlinear reaction-diffusion equation problem, we use the prefix $P$ to indicate the use of

Table 2: Ablation study for 1D singular perturbation

| Neural network structures | Relative $l^2$ error |
|---|---|
| $[50], [100]$ | 31.36% |
| $[50], [100], [100]*2$ | 11.05% |
| $[50], [100], [100]*2, [100]*3$ | 10.79% |
| $[50], [100], [100]*2, [100]*3, [100]*2, F_{10}[50]*2$ | 0.85% |
| $[50], [100], [100]*2, F_{10}[100]*3$ | 1.1% |
| $[512]*6$ | 15.16% |
| $F_{10}[256]*4$ | 0.60% |
| $F_{10}[50]*2$ | 1.27% |
| $[50], [100], [100]*2, [100]*3, F_{10}[50]*2$ | 0.43% |

Table 3: Ablation study for 2D singular perturbation with boundary layers

| Neural network structures | Relative $l^2$ error |
|---|---|
| $[50], [100]$ | 61.56% |
| $[50], [100], [100]*2$ | 57.01% |
| $[50], [100], [100]*2, [100]*3$ | 57.25% |
| $[50], [100], [100]*2, [100]*3, [100]*2$ | 57.23% |
| $[50], [100], [100]*2, F_{50}[100]*2$ | 2.67% |
| $F_{50}[256]*4$ | 4.01% |
| $F_{50}[100]*2$ | 15.56% |
| $[50], [100], [100]*2, [100]*3, [100]*2, F_{50}[100]*2$ | 1.03% |

a periodic fully-connected neural network. The weight for each study is set to $2^{-n}$, where $n$ is the index of the neural network.

In order to facilitate clear comparisons, the reported results will be presented in the last line of the following tables.

In the following results, it becomes evident that incorporating a Fourier feature network enhances performance, primarily by smoothing the neural tangent kernels. However, we argue that relying solely on a single Fourier feature network is insufficient for attaining the level of accuracy achieved by GB PINNs. As corroborated by data presented in Table 2 through Table 4, specifically in the penultimate row, there exists a discernible gap in accuracy between a standalone Fourier feature network and GB PINNs. This discrepancy is particularly noticeable in 2D problems. Therefore, the superior accuracy of GB PINNs can be attributed to the synergistic interaction among multiple networks, rather than the implementation of a Fourier feature network alone.

Table 4: Ablation study for 2D singular perturbation with an interior boundary layer

| Neural network structures | Relative $l^2$ error |
|---|---|
| $[200]*3, [100]*3, [100]*2$ | 8.69% |
| $[200]*3, [100]*3, F_5[50]*2$ | 6.46% |
| $[200]*3, [100]*3, [100]*2, F_5[50]*2, [100]*3$ | 6.03% |
| $F_5[512]*4$ | 38.9% |
| $F_5[50]*2$ | 16.15% |
| $[200]*3, [100]*3, [100]*2, F_5[50]*2$ | 3.37% |

Table 5: Ablation study for 2D nonlinear reaction-diffusion equation

| Neural network structures | Relative $l^2$ error |
|---|---|
| $P[200] * 3$ | 1.18% |
| $P[200] * 3, P[100] * 3$ | 1.18% |
| $P[200] * 3, P[100] * 3, P[100] * 2, P[100] * 3$ | 0.59% |
| $P[200] * 3, P[100] * 3, P[100] * 2$ | 0.58% |

