# OpenReview forum: "Ensemble learning for Physics Informed Neural Networks: a Gradient Boosting approach"
_ICLR.cc/2024/Workshop/AI4DiffEqtnsInSci — AI4DiffEqtnsInSci @ ICLR 2024 Poster_

### Official Review · Reviewer_hhah · 2024-02-27
**Gradient boosting for PINNs shows order of magnitude performance improvement over vanilla PINNs**

**Rating:** 5
**Confidence:** 3

**Review:**

Authors present a boosting technique to train PINNs. They focus on PDE problems where vanilla PINNs show bad accuracies and show that their method provides order of magnitude performance improvement at the cost of 3x training time and 2x memory time.

Strengths:
1. Paper is clear and well-written
2. GB-PINNs show highest accuracy for all systems considered making it a competitive method.

Weaknesses:
Probably the biggest weakness of this paper is the poor baselines and the ablations w.r.t fourier features.

1. Comparison to vanilla PINNs may be unfair at this point. Vanilla PINNs are known to have many issues and there have been many papers trying to address them. Can the authors comment on their methodology compared to the many variants of PINNs currently present?
2. Similar to 1., the accuracy comparisons should probably also include vanilla PINNs of similar training and mem costs. So, maybe more collocation points/bigger models for the vanilla PINN. This would help the reader understand that given x resources, what is the accuracy gain. Is it still O(10) times or does it become O(1).
   - this ties back to 1: would sequence-to-sequence in 3.4 (which also introduces more cost by making the training sequential) or parameter continuation (for other systems, could start with some larger \eps and slowly decay it) be competitive alternatives as well? It's more useful to the reader to show these baselines rather than vanilla PINN.
3. In many real scientific systems, physics-based losses are only a part of the process. Supervision from observational data and/or simulations typically guide the neural network to better solutions. I'm wondering how much of the vanilla PINN failure modes can simply be fixed by a few data points (supervised loss) at minimal cost?
4. The PINN errors seem really large. Does the training fail/plateau? This again ties back to 2. and 1. It seems that the authors are primarily trying to fix the optimization difficulties. Other baselines here would be more beneficial than vanilla PINN which is known to be a poor baseline for many PDEs today.
5. In 3.4 seems like the vanilla PINN (Krishnapriyan et al. reference) doesnt use fourier features. Is that right? In which case, the vanilla PINN baseline should include it since the inputs are 0,2\pi which are bad input ranges for NNs in general.
6. Not sure if I'm misunderstanding the ablations: but it seems like the fourier embeddings substantially improve the accuracy on their own. For each system vanilla PINN + fourier features gives the following accuracies: 0.6%, 4%, 16% and 1%. If that is true, this weakens the contributions of GB-PINNs for all systems except system 3. This needs to be emphasized in the main text, because (similar to 2.) the reader needs to be clear on what the best method is given a computational/memory envelope and fourier features add no cost. Please comment.

For reasons highlighted in the weaknesses, I lean towards 5. but open to raising my score based on author's comments/other reviewers or if I misunderstood some ablations.

---

### Official Review · Reviewer_S1Y6 · 2024-02-27
**solid work, fitting the workshop's topic**

**Rating:** 7
**Confidence:** 3

**Review:**

The reviewed work investigates the use of gradient boosting for improving the training of physics-informed neural networks. Instead of training a single neural network to fit the solutions, the authors propose using a sequence of neural networks, each of which trained on the residual of the approximation based on the previous networks.

The authors find that this approach can significantly improve the accuracy of the resulting approximate solution on problems with multi-scale structure.

The paper is well-written and appears to achieve significant improvements in the accuracy of PINN training on multiscale problems - one of their current weaknesses. I therefore recommend acceptance.

---

### Meta-Review · Area_Chair_n635 · 2024-02-28

**Recommendation:** Accept (Poster)

**Metareview:**

Dear Authors,

Thank you for submitting the draft.

Both reviewers agree that the presented results are competitive and the draft is clearly written. However, Reviewer hhah does also raise some major points of concern. It is expected that authors will be addressing comments by the reviewers in the final draft.

regards

AC

---

### Decision · Program_Chairs · 2024-02-29

Accept (Poster)